# Plasma neurofilament light protein correlates with diffusion tensor imaging metrics in frontotemporal dementia

**Nicola Spotorno**[1,2]*, **Olof Lindberg**[3], **Christer Nilsson**[4], **Maria Landqvist Waldö**[5], **Danielle van Westen**[6], **Karin Nilsson**[2], **Susanna Vestberg**[7], **Elisabet Englund**[8], **Henrik Zetterberg**[9,10,11,12], **Kaj Blennow**[9,10], **Jimmy Lätt**[13], **Nilsson Markus**[6], **Wahlund Lars-Olof**[3], **Santillo Alexander**[2]

**1** Department of Neurology, Penn Frontotemporal Degeneration Center, University of Pennsylvania Perelman School of Medicine, Philadelphia, PA, United States of America, **2** Department of Clinical Sciences, Clinical Memory Research Unit, Lund University, Malmö, Sweden, **3** Division of Clinical Geriatrics, Karolinska Institute, Stockholm, Sweden, **4** Division of Neurology, Department of Clinical Sciences, Lund University, Lund, Sweden, **5** Department of clinical Sciences, Clinical Sciences Helsingborg, Lund, Lund University, Lund, Sweden, **6** Department of Diagnostic Radiology, Clinical Sciences, Lund University, Lund, Sweden, **7** Department of Psychology, Lund University, Lund, Sweden, **8** Division of Pathology, Department of Clinical Sciences, Lund, Sweden, **9** Department of Psychiatry and Neurochemistry, Institute of Neuroscience & Physiology, the Sahlgrenska Academy at the University of Gothenburg, Mölndal, Sweden, **10** Clinical Neurochemistry Laboratory, Sahlgrenska University Hospital, Mölndal, Sweden, **11** Department of Neurodegenerative Disease, UCL Institute of Neurology, Queen Square, London, United Kingdom, **12** UK Dementia Research Institute at UCL, London, United Kingdom, **13** Center for Medical Imaging and Physiology, Skåne University Hospital, Lund, Sweden

* nicola.spotorno@pennmedicine.upenn.edu

**Data Availability Statement:** Data are available upon request. Data contain sensitive information about patients status. We are prohibited from

## Abstract

Neurofilaments are structural components of neurons and are particularly abundant in highly myelinated axons. The levels of neurofilament light chain (NfL) in both cerebrospinal fluid (CSF) and plasma have been related to degeneration in several neurodegenerative conditions including frontotemporal dementia (FTD) and NfL is currently considered as the most promising diagnostic and prognostic fluid biomarker in FTD. Although the location and function of filaments in the healthy nervous system suggests a link between increased NfL and white matter degeneration, such a claim has not been fully elucidated *in vivo*, especially in the context of FTD. The present study provides evidence of an association between the plasma levels of NfL and white matter involvement in behavioral variant FTD (bvFTD) by relating plasma concentration of NfL to diffusion tensor imaging (DTI) metrics in a group of 20 bvFTD patients. The results of both voxel-wise and tract specific analysis showed that increased plasma NfL concentration is associated with a reduction in fractional anisotropy (FA) in a widespread set of white matter tracts including the superior longitudinal fasciculus, the fronto-occipital fasciculus the anterior thalamic radiation and the dorsal cingulum bundle. Plasma NfL concentration also correlated with cortical thinning in a portion of the right medial prefrontal cortex and of the right lateral orbitofrontal cortex. These results support the hypothesis that blood NfL levels reflect the global level of neurodegeneration in bvFTD and help to advance our understanding of the association between this blood biomarker for FTD and the disease process.

sharing the data publicly for general research that was not described in the consent procedure. Data are available upon request from researchers who have ethical approval from the Skåne University Hospital, Bild och Funktion, forskningssamarbete. bf.sus@skane.se. Further data access queries can be directed to Sophia Zackrisson, Skånes universitetssjukhus, Diarium, Rådhus Skåne, 291 89 Kristianstad, Sweden, e-mail address: sophia. zackrisson@skane.se, phone: +46 40 33 87 97

**Funding:** The funders had no role in study design, data collection and analysis, decision to publish, or preparation of the manuscript. NS, OL, LOW and AS were all supported by the Schörling foundation. NS is also supported by the National Institute on Aging (NIA), grant numbers: AG052943, AG017586. AS is also supported by the Swedish Society for Medical Research, The Benthe Rexhed Gersteds foundation, The Bundy Academy, The Strategic Research Area MultiPark (Multidisciplinary Research in Parkinson's disease) at Lund University, and Region Skåne. HZ is a Wallenberg Scholar supported by grants from the Swedish Research Council (#2018-02532), the European Research Council (#681712), Swedish State Support for Clinical Research (#ALFGBG-720931), the Alzheimer Drug Discovery Foundation (ADDF), USA (#201809-2016862), and the UK Dementia Research Institute at UCL. KB is supported by the Swedish Research Council (#2017-00915), the Alzheimer Drug Discovery Foundation (ADDF), USA (#RDAPB-201809-2016615), the Swedish Alzheimer Foundation (#AF-742881), Hjärnfonden, Sweden (#FO2017-0243), the Swedish state under the agreement between the Swedish government and the County Councils, the ALF-agreement (#ALFGBG-715986), and European Union Joint Program for Neurodegenerative Disorders (JPND2019-466-236).

**Competing interests:** HZ has served at scientific advisory boards for Denali, Roche Diagnostics, Wave, Samumed and CogRx, has given lectures in symposia sponsored by Fujirebio, Alzecure and Biogen, and is a co-founder of Brain Biomarker Solutions in Gothenburg AB (BBS), which is a part of the GU Ventures Incubator Program. KB has served as a consultant, at advisory boards, or at data monitoring committees for Abcam, Axon, Biogen, JOMDD/Shimadzu. Julius Clinical, Lilly, MagQu, Novartis, Roche Diagnostics, and Siemens Healthineers, and is also a co-founder of Brain Biomarker Solutions in Gothenburg AB (BBS). These do not alter our adherence to PLOS ONE policies on sharing data and materials.

# Introduction

In the central nervous system, neurofilaments are cytoskeletal components of neurons that are particularly abundant in axons, where neurofilaments provide structural support and contribute maintaining size, shape, and caliber of the axons [1]. Neurofilament light chain (NfL) is the most abundant and soluble of the neurofilament subunits and can be reliably measured in cerebrospinal fluid (CSF) as well as in blood. The high correlation between CSF and blood concentrations of NfL (e.g., in plasma [2,3]) opened the avenue to the development of an accurate and noninvasive biomarker for diagnostics and, possibly, for monitoring the effect of disease modifying drugs during treatment trials. In recent years, NfL has rapidly emerged as one of the most promising fluid biomarkers in several neurodegenerative conditions including Alzheimer's disease [4] and atypical parkinsonian disorders [5]. Early on a markedly increased CSF levels of NfL in frontotemporal dementia (FTD) as compared with controls and Alzheimer´s disease was found [6], a finding valid also for NfL in blood [7], and increased peripheral NfL has now been reliably validated in FTD [8–13]. A dramatic increase compared with controls and other neurodegenerative conditions has been also exhaustively documented in amyotrophic lateral sclerosis (ALS) [14–16].

NfL is mostly found in long, myelinated axons, and was found early to correlate with white matter changes (white matter is, together with myelin and glia, constituted by axons) [17], but only few studies have investigated the association between NfL and other in vivo markers that reflect the involvement of white matter in the neurodegenerative process [12,18]. In ALS, increased NfL concentration in both CSF and blood have been reported to correlate specifically with the degeneration of the cortico-spinal tract [16,19], while in FTD, both Steinacker and colleagues [12] and Sherling and colleagues [18], have shown an association between the serum concentration of NfL and decrease in frontal white matter volume. Understanding the link between peripheral NfL and aspects of the neurodegenerative process in FTD would be critical to guide the interpretation of NfL levels in the context of both clinical practice and treatment trials. To the best of our knowledge, no previous study has investigated the link between peripheral NfL level and the pathological changes in white matter occurring in FTD using diffusion tensor imaging (DTI) which is one of the most sensitive imaging techniques to capture such changes *in vivo*.

In the present study, we used DTI to explore the potential relationship between plasma NfL concentration and white matter degeneration in a group of patients affected by behavioral variant FTD (bvFTD). DTI has already proved to be able to reflect white matter degeneration profiles in FTD [20] and considering the results of previous studies, we aimed to test whether the peripheral concentration of NfL are associated with the degeneration of specific tracts, as appears to be the case in ALS, or if NfL reflects the general level of severity of white matter atrophy in bvFTD, as suggested by the previous studies [12,18]. To this end, the association between DTI metrics and NfL levels was investigated both unbiasedly at the whole-brain white matter level and in a region-of-interest analysis focusing on white matter tracts that are known to be involved in the bvFTD pathological process. Previous studies on FTD based on the quantification of DTI metrics [20–22] as well as on a combination of tensor based tractography and quantification of DTI metrics [23] have indeed showed extensive pattern of white matter involvement including the fronto-occipital fasciculus, the superior longitudinal fasciculus, the uncinate fasciculus, the dorsal cingulum bundle, the anterior thalamic radiation, and the inferior/hippocampal portion of the cingulum bundle. We also included in our analysis tracts in which degeneration has been reported to correlate with NfL level in previous DTI studies, in particular the cortico-spinal tract [16,19]. Moreover, we aimed to explore the potential association between NfL levels and cortical thickness to investigate the specificity of the hypothesized link between plasma NfL levels and white matter degeneration.

## Material and methods

### Participants

Twenty patients affected by the behavioral variant frontotemporal dementia (bvFTD) were included in the study. The participants took part in the Lund Prospective Frontotemporal Dementia Study (LUPROFS), a longitudinal study of patients with any of the frontotemporal dementia spectrum disorders which included patients from 2009 to 2014 at the Memory Clinic of Skåne University Hospital in Lund, Sweden. The LUPROFS protocol includes clinical examination, caregiver history, symptom rating, neuropsychological examination, standardized neurological examination and, at baseline, CSF and blood sample and MRI. bvFTD patients were diagnosed by a multidisciplinary team accordingly to the International Behavioral Variant FTD Consortium Criteria [24]. Genetic screening for mutations in the genes of microtubule-associated protein tau (MAPT), progranulin (GRN), and in chromosome 9 open reading frame 72 (C9ORF72) was performed in all patients. Postmortem neuropathological examination was aimed for in all patients deceased during follow up. The exclusion criteria included: > 3 lacunar strokes or any number of other type of strokes visible on MRI examination, alcohol addiction, or any other significant neurological or psychiatric comorbidity. Patients were included in the present study only if they had a clinical diagnosis of probable or definite bvFTD, underwent MRI successfully with all relevant sequences to this study, and underwent MRI within a year from blood sampling. Twenty of the 41 subjects with bvFTD included in the LUPROFS cohort, could be included in the present study. No standardized assessment of motor functions was performed as part of the study, however, one patient had a diagnosis of FTD with motor neuron disease (FTD-MND, according to the Awji criteria [25]). More details on inclusion procedure and instruments can be found in [23]. 22 healthy controls were also included in the study for the purpose of building an unbiased white matter template (see the sction on the Diffusion tensor imaging (DTI) analysis). Healthy controls underwent the same protocol as the included patients. Demographic and clinical features of the cohort are presented in Table 1. The study was approved by the Regional Ethical Review Board, Lund, Sweden (Number 617/2008). Patients and healthy controls were informed of the study content in both oral and written form. Informed consent was obtained in written form.

**Table 1. Demographic information.**

|  | bvFTD | Healthy controls |
|---|---|---|
| Number (Female) | 20 (10) | 22 (11) |
| Diagnostic confidence* | 9 cases–definite[+] 11 cases–probable | |
| Age (std.) | 68 (9) | 65 (11) |
| MMSE (std.) | 23.2 (4.9) | 29.6 (0.5) |
| Interval plasma–MRI in months (std; interval range in months) | 0.2 (3.1; 0–10) | – |
| FTLD-CDR (std.) | 1.7 (0.6) | |
| FTLD-CDR sum of boxes (std.) | 8.7 (4.0) | – |

Values are given as mean and standard deviation (std).

Abbreviations: bvFTD: behavioural variant frontotemporal dementia; FTLD-CDR = frontotemporal lobar degeneration Clinical Dementia Rating; MMSE: Mini Mental State Examination.

*According to International Behavioral Variant FTD Consortium Criteria [24]. [+] 6 patients fulfill definite bvFTD criteria according to neuropathological examination, 2 according to genetic examination, and one according to both. Demographic factors and clinical characteristics were compared using Chi-square ($x^2$) and Mann-Whitney U tests [Sex: $x^2 = 0$, p > 0.9; Age: U = 155, p = 0.052; MMSE: U = 13.5, p < 0.001].

## Imaging protocol

MRI was performed using a Philips Achieva 3T scanner equipped with an eight-channel head coil. Diffusion weighted imaging (DWI) was performed with an echo-planar single-shot spin echo sequence. Diffusion encoding was performed in 48 directions at a b-value of 800 s/mm$^2$ along with 1 volume with b-value of 0. The voxel size was 2×2×2 mm$^3$, TR 7881 ms and TE 90 ms. A T1-weighted 3D volumetric sequence was also acquired with a voxel size of 1×1×1 mm$^3$, TR 8.3 ms, TE 3.84 ms, FOV 256×256×175 mm$^3$.

## Plasma neurofilament light chain (NfL) protocol

Plasma NfL concentration was measured using an in house ultrasensitive enzyme-linked immunosorbent assay on a Single molecule array platform (Quanterix Corp, Billerica, MA, USA), as previously described [26]. All measurements were performed in one round of experiments, using one batch of reagents with intra-assay coefficients of variation below 10%.

## Diffusion tensor imaging (DTI) analysis

The diffusion data were corrected for motion and eddy current induced artifacts using an extrapolation-based registration approach as previously reported [27]. A single diffusion tensor model was fitted to these data using FMRIB Software Library and parametric maps of fractional anisotropy (FA), mean diffusivity (MD), radial diffusivity (RD), and axial diffusivity (AD) were subsequently computed. Tract-based spatial statistics pipeline (TBSS, [28]) was then applied to the parametric maps. In brief, the FA maps were warped to the FMRIB58_FA standard template (FMRIB, University of Oxford, UK; resolution: 1×1×1mm$^3$) in MNI152 space (Montreal Neurological Institute, McGill University, Canada) using FMRIB's non-linear registration tool (FNIRT v1.0). All the warped FA maps, including the maps of the healthy controls, were subsequently averaged to create a mean FA template, from which the FA skeleton was computed, imposing an FA threshold of 0.2. The FA maps of each participant were then, projected onto the skeleton as well as the other DTI scalars. In the present study we focus especially on a subset of tracts of interest, that consistently have been reported as affected in bvFTD in previous DTI studies [20,21,23] namely: the fronto-occipital fasciculus (FOF), the superior longitudinal fasciculus (SLF), the uncinate fasciculus (UF), cortico-spinal tract (CST), the dorsal cingulum bundle (dCin), the anterior thalamic radiation (ATR), and the inferior/ hippocampal portion of the cingulum bundle (iCin). In order to extract summary metric from these tracts the Johns Hopkins University white-matter tractography atlas (available in FMRIB software library; FSL) [29] were intercepted with study-wise FA skeleton and the median values of FA, MD, RD and AD were extracted from each tract of interest for each participant.

## Cortical thickness analysis

Cortical reconstruction and volumetric segmentation were performed on the T1-weighted 3D image using Freesurfer 5.3 image analysis pipeline which is documented and freely available for download online (http://surfer.nmr.mgh.harvard.edu/). The technical details of these procedures are described in prior publications, which are listed at https://surfer.nmr.mgh.harvard.edu/fswiki/FreeSurferMethodsCitation. Briefly, the whole-brain T1-weighted images underwent a correction for intensity homogeneity, skull striping, and segmentation into grey and white matter with intensity gradient and connectivity among voxels. Cortical thickness was measured as the distance from the gray/white matter boundary to the corresponding pial surface. Assessment of intracranial volume was also performed using Freesurfer.

Reconstructed data sets were visually inspected for accuracy, and segmentation errors were corrected. Quality control of the Freesurfer output was performed by OL.

### Statistical analysis

A set of whole-skeleton voxel-wise regression analyses on the DTI metrics was performed with the Randomize tool [30] version 2.9 available in FSL with 10000 permutations, threshold-free cluster enhancement [31] and 2D optimization for tract-based DTI analysis. NfL level was included as independent variable in each model along with age at scan as a nuissance covariate. Whole-skeleton statistical significance was set at the family-wise error corrected threshold of $p<0.05$.

Potential differences between DTI metrics in the tracts of interest across hemisphere were assessed with Wilcoxon test. The association between DTI metrics and NfL was investigated employing multiple linear least square regression models and Benjamin Hochberg procedure for False Discovery Rate (FDR; p-value threshold = 0.05). Plasma NfL levels were log transformed and modeled as independent variable along with age as a nuisance covariate.

Cortical thickness analysis was performed using the QueryDesign Estimate Contrast (QDEC) tool. In parallel with the DTI analysis the model includes cortical thickness as dependent variable while plasma NfL levels were modeled as independent variable along with age at time of MRI scan as nuisance covariate. The results of the GLM analysis were corrected for multiple comparisons at the cluster level using the Monte Carlo simulation. The level of statistical significance was evaluated using a cluster-wise P (CWP) value correction procedure for multiple comparisons based on a Monte Carlo z-field simulation with a cluster forming vertex-z-threshold of 1.3 corresponding to $p < 0.05$. All the analyses were performed using Python (3.7) except the voxel-wise analyses (as previously described).

## Results

### Voxel-wise TBSS analysis

The voxel-wise regression analysis in patients affected by bvFTD revealed a negative correlation between plasma NfL levels and a widespread pattern of reduction in FA values which encompass the uncinate fasciculus bilaterally, the anterior thalamic radiation, the corpus callosum (genu, body and splenium), the inferior fronto-occipital fasciculus (more extensively on the right side) and the left corticospinal tract/cerebral peduncle (see Fig 1). No positive correlation between plasma NfL levels and FA was found and plasma NfL levels did not correlate with any other of the DTI metrics (MD, RD or AD) after correction for multiple comparisons.

### Tracts of interest analysis

Considering that the voxel-wise analysis revealed significant results only for FA we focused the tract of interest analysis on this metric. The Wilcoxon test showed significant differences in FA values between the right and left side of some of the tracts of interest [FOF: W = 170, $p<0.001$; dCin: W = 247, $p<0.05$; SLF: W = 281, $p<0.05$]. Therefore, further analysis was performed separately for the two hemispheres.

A negative correlation between FA and plasma NfL level was significant after correction for multiple comparisons (Bonferroni corrected alpha = 0.05/12) in several tracts of interests namely in the FOF [lFOF: β = -0.03, FDR-$p<0.05$; rFOF: β = -0.03, FDR-$p<0.05$], in the SLF [lSLF: β = -0.03, FDR-$p<0.05$; rSLF: β = -0.03, FDR-$p<0.05$], in the left UF [β = -0.03, FDR-$p < 0.05$], in the right ATR [β = -0.04, FDR-$p<0.05$] and in the right dCin [β = -0.04, FDR-$p<0.05$]. The correlation between FA values and NfL in the left ATR, the right UF and in the left dCin was only marginally significant [lATR: β = -0.02, p = 0.06; rUF: β = -0.04, p = 0.06;

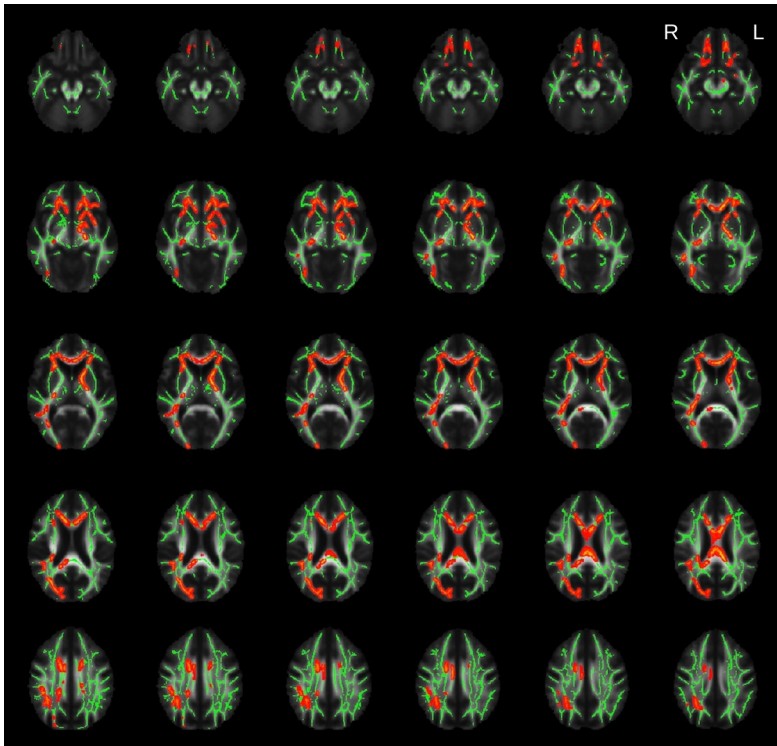

**Fig 1. Results of the voxel-wise TBSS analysis.** Highlighted clusters (in red) represent significant (family wise error, FWE < 0.05) age corrected negative correlations between fractional anisotropy (FA) and plasma neurofilament light (NfL) protein levels among the patients with behavioural variant frontotemporal dementia (bvFTD). The average skeleton is displayed in green. The results were overlaid onto the study-specific mean FA map.

ldCin: β = -0.03, p = 0.05]. No significant correlation was found in the CST [lCST–group: β = -0.01, p>0.1; rCST–group: β = -0.01, p>0.1] (see Fig 2 and S2 Appendix for the complete statistical results). FA was significantly reduced in the bvFTD group relative to the healthy controls in all the tracts of interest except in the inferior cingulum bundle (see S1 Appendix).

## Cortical thickness analysis

Whole-brain regression analysis revealed a significant negative correlation between cortical thickness and plasma NfL level in the right medial prefrontal cortex and in the right lateral orbitofrontal cortex (see Fig 3).

## Discussion

This study explored for the first time the association between plasma NfL levels and DTI metrics in a group of patients affected by bvFTD. We first took a voxel-wise approach to test the extent of the association between DTI metrics and NfL across all white matter structures and then we focused specifically on a subset of tracts of interest, that consistently have been reported to be affected in previous DTI studies on bvFTD [20–22]. The voxel-wise analysis revealed a pattern of negative association between FA and plasma NfL levels that was consistent with the spatial distribution of white matter degeneration in bvFTD as assessed by neuropathology [32] and previous DTI studies [20–22]. The analysis of specific tracts of interest also showed a negative correlation between plasma NfL levels and FA, in all the tracts under examination with the only exception of the cortico-spinal tract. These results suggested a strong link

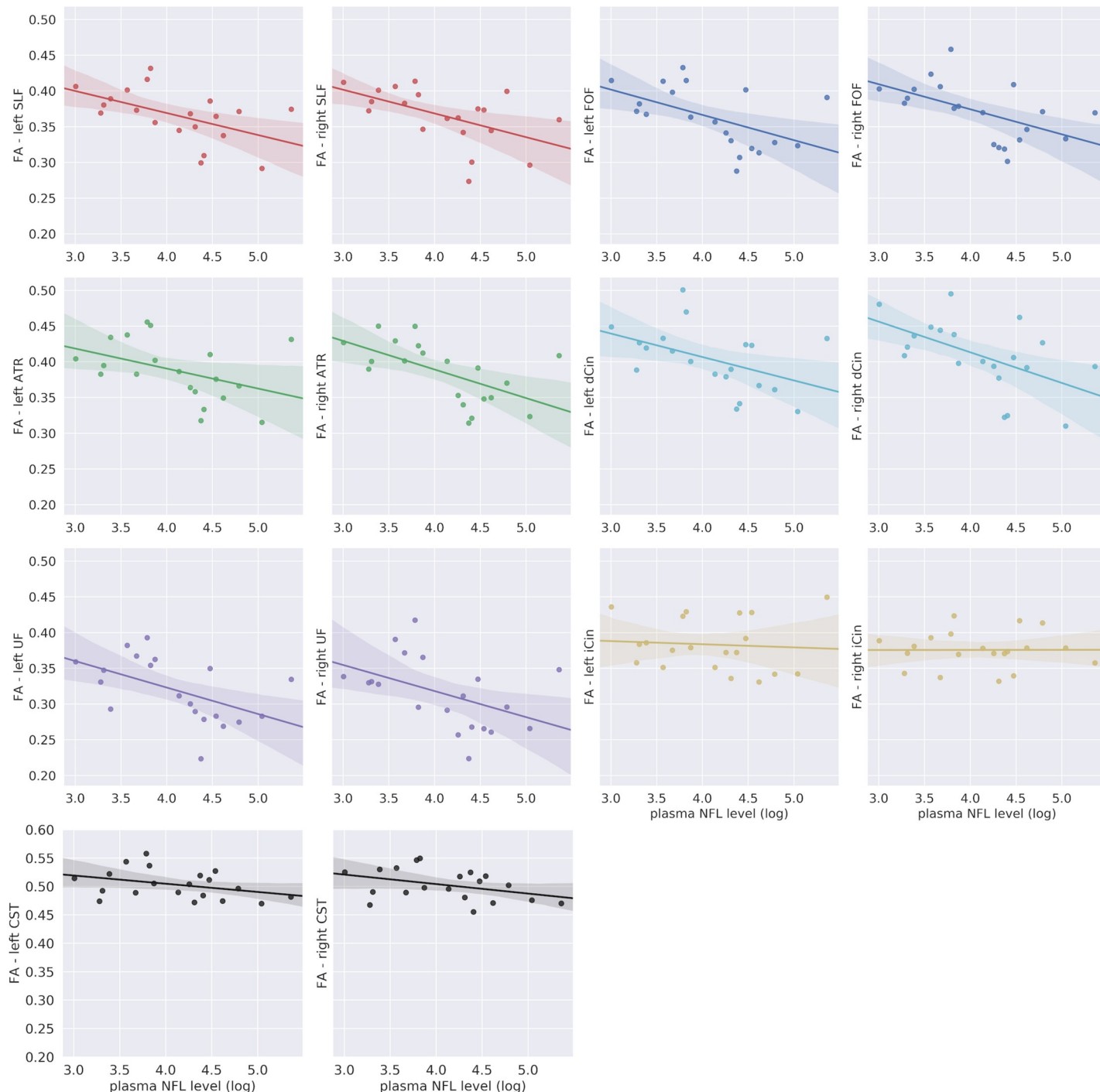

**Fig 2. Regional analysis of specific tracts.** Abbreviations: ATR: anterior thalamic radiation; CST: corticospinal tract; dCin: dorsal cingulum bundle; FA: fractional anisotropy; FOF: fronto-occipital fasciculus; iCin: inferior/hippocampal portion cingulum bundle; NfL: neurofilament light chain; SFL: superior longitudinal fasciculus; UF: uncinate fasciculus. Median FA values for each tract of interest plotted as a function of the log transformed plasma NfL levels among the patients with behavioral variant frontotemporal dementia bvFTD. The translucent area around the regression line represents the 95% confidential interval for the regression estimate.

between white matter involvement in bvFTD and the increase of plasma NfL level, providing further evidence of the link between peripheral NfL concentration and the central pathological process in bvFTD.

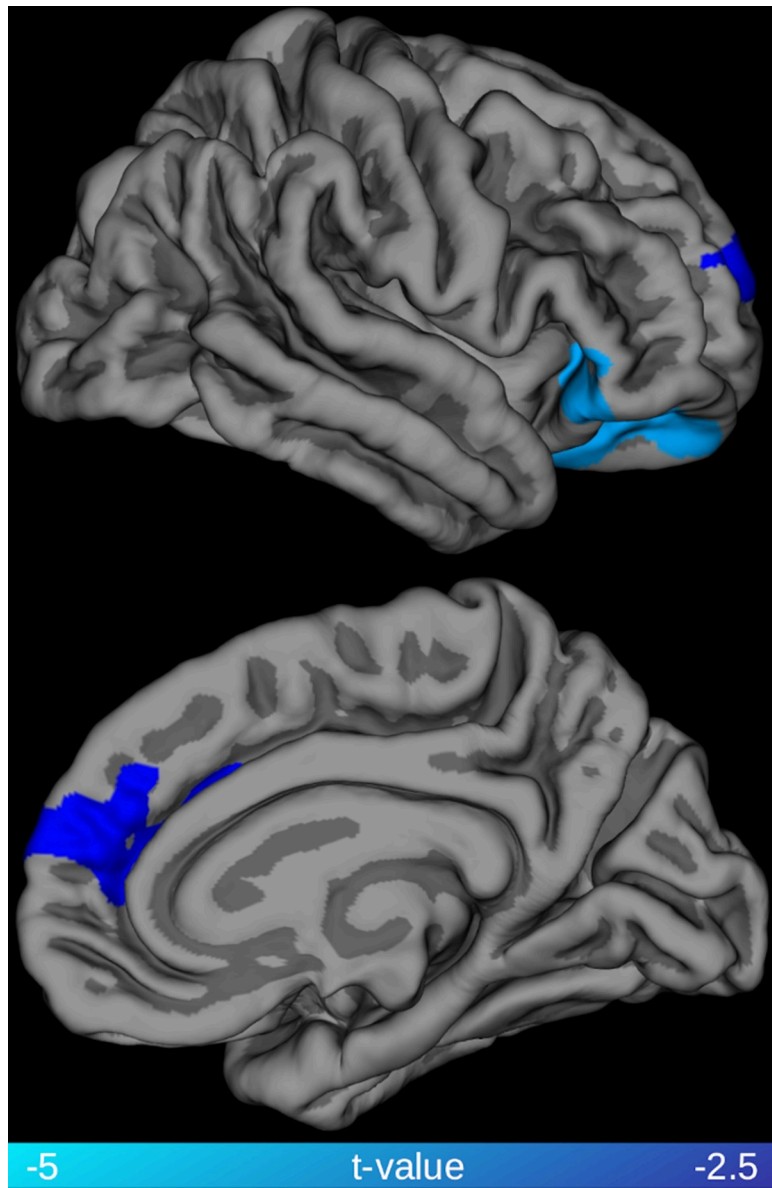

**Fig 3. Relationship between cortical thickness and plasma NfL levels.** Highlighted clusters represent significant (family-wise error, FWE < 0.05) age corrected negative correlations between cortical thickness and plasma neurofilament light chain (NfL) levels among the patients with behavioral variant frontotemporal dementia (bvFTD).

NfL is the most promising fluid biomarker for diagnostics and monitoring degeneration in FTD both in the context of clinical practice as well as in clinical trials [33] In the diagnostic process, NfL is the only fluid biomarker with potential to separate bvFTD from psychiatric disorders [34,35], which constitute the main differential diagnosis for bvFTD [36]. Plasma is arguably preferable to CSF for NfL measurements due to the minimally invasive procedure required to sample it. Considering the potential value of this marker, it is relevant to clarify what aspect of the FTD pathological process is reflected by NfL levels. The present results point to a clear link between white matter degeneration (and presumably neuronal axons, its main constituent together with myelin and glia) and NfL (presumably mainly emanating from neuronal axons) levels in plasma. Both CSF and blood concentrations of NfL have been

previously associated with *grey* matter atrophy in several neurodegenerative diseases, including FTD. For example, van der Ende and colleagues have recently reported that the rate of changes in serum NfL concentration in genetic FTD cases was correlated with rate of volume reduction in several cortical and subcortical regions including frontal lobe, insula, cingulate gyrus, hippocampus, putamen, temporal lobe and amygdala [37]. Similarly, Meeter et al. have shown a negative association between NfL levels in the CSF and parahippocampal volume in a large cross-sectional study on the semantic variant of primary progressive aphasia [9]. However, these studies did not assess the association between NfL and white matter involvement. Both Scherling and colleagues and Steinacker and colleagues have, instead, reported negative correlation between CSF concentration of NfL and both gray and white matter volume across different phenotypes of FTD [18]. The present study using DTI metrics indicate a spatially more diffuse relationship between NfL and white matter, compared to the association between NfL levels and white matter volume. Although, a direct comparison between studies employing alternative techniques is not possible, the difference could be due to the superior sensitivity of DTI in detecting pathological processes in white matter. Although NfL are structural components of axons, they are also present in gray matter; therefore, gray matter degeneration could contribute to the increase of peripheral NfL concentration. In Alzheimer's disease, plasma NfL levels have been shown to correlate both cross-sectionally and longitudinally with cortical thinning in regions typically affected by the Alzheimer neurodegenerative process, such as entorhinal, inferior temporal, middle temporal, and fusiform cortex [4]. In the present study, plasma NfL levels were also associated with cortical thinning in a portion of the right medial prefrontal cortex and the right lateral orbitofrontal cortex. These regions constitute a subset of the prefrontal, insular and temporal cortex typically affected by atrophy in bvFTD [38]. The extent of the association between NfL and cortical thinning appears to be relatively modest compared to the widespread pattern of associations with FA. Such a difference could be related to the biology of NfL, which are particularly abundant in highly myelinated axons, but the difference in statistical results could also be due to lack of power for detecting statistical associations with a morphometric analysis.

In the present study we tested whether bvFTD plasma NfL levels are associated with a particular spatial profile of white matter involvement, where several scenarios could be hypothesized. In ALS, an increase in the peripheral concentration of NfL levels has been associated with changes in diffusion metrics mostly in the cortico-spinal tracts [14–16]. Thus, given the close relationship between ALS and FTD, a subclinical corticospinal tract involvement could drive the increased NfL levels in FTD. Analogously, a semantic dementia (i.e. semantic variant of primary progressive aphasia) like pattern, with NfL correlations with the uncinate fasciculus mainly, could be expected given that semantic dementia shows higher NfL levels than other primary progressive aphasias and has a characteristic involvement of the uncinate fasciculus [9]. A specific association with the uncinate fasciculus could also be expected given that this is one of the earliest tracts affected in bvFTD [39] and increased NfL seem to be an early event in FTD [37]. Both the voxel-wise results and the regional analysis seems to point, instead, to a diffuse association of NfL with FA in the main tracts that are involved in the neurodegenerative process of bvFTD. The association appeared to be prominent in the majority of tracts studied, which all are long interlobar tracts with diverse functions (i.e. association, commissural, and projection tracts), with the notable exception of the CST. The results in the CST appear counter intuitive given what is reported above on regarding ALS. However, there are inconsistencies and the largest study of ALS to date has indeed not shown correlation between NfL and DTI metrics [40].

The association between plasma NfL levels and FA values in the majority of tracts affected by the disease process is in line with both cross-sectional and longitudinal studies in the FTD spectrum showing that levels of NfL are increased with disease severity and associated with

poorer prognosis [7,10,11,18,37,41]. Although it is not the primary outcome of the present study, the association between NfL levels and disease severity holds also in our cohort has revealed by a positive association between NfL and FTLD-CDR sum of boxes [Spearman's rho = 0.7; p < 0.001]. Of particular interest is that both Meeter and colleagues [11] and van der Ende and colleagues [37] showed that in genetic FTD cases, the serum concentration of NfL is clearly higher in symptomatic versus asymptomatic mutation carriers. We believe the current finding further strengthen the validity of NfL as a biomarker of disease severity and thus potentially as a surrogate outcome measure in treatment trials. What emerges as a caveat from this, and previous studies, is that the association with disease severity could mean that NfL is less useful in patients that not yet have a full clinical bvFTD dementia (i.e. prodromal bvFTD, or bvFTD at the mild cognitive impairment level), which would limit its application in the psychiatric differential diagnostic setting.

In addition to the relatively small sample size, other limitations should be taken into account while evaluating the results of the present study. bvFTD is a clinical phenotype, and, currently there is no in vivo marker that can disentangle the underlying pathology, namely tauopathy or TAR DNA-binding protein 43 (TDP-43) proteinopathy. Evidence of increased NfL levels in ALS/MND, which is most commonly a TDP-43 proteinopathy [42–44], and in the semantic variant primary progressive aphasia [9,10], which also most commonly characterized by misfolding of TDP-43 [45], suggest a link between NfL and TDP-43 proteinopathy in the FTD spectrum [10]. However, other studies have shown that NfL levels are elevated in primary tauopathies like progressive supranuclear palsy [5]. Although, the purpose of our study was the investigation of the association between plasma NfL levels and DTI metric in a clinical phenotype (bvFTD), the potential confounding effect of mixed underlying proteinopathies must be taken into consideration. Increased peripheral NfL has now been reliably validated in FTD [8–13] and was not a purpose of the current study; however the lack of a direct comparison of plasma NfL levels between bvFTD and healthy controls is also a limitation of the present study. Moreover, the lack of association between plasma NfL levels and DTI metrics other than FA in the TBSS analysis is surprising when considering the extent of the association between NfL levels and FA values. A follow up study in a bigger cohort is warranted to investigate whether, with greater statistical power, associations with other metrics would emerge. Different DTI metrics could also be differently sensitive to different biological process. For example, FA appears to distinguish between progressive supranuclear palsy and Parkinson's disease with higher level of accuracy than other DTI metrics [46], while axial and mean diffusivity seem to be the first DTI-related abnormalities to occur in Alzheimer's disease [47]. However, one should bear in mind that, although DTI metrics are sensitive to pathological processes in white matter, they cannot be interpreted as a direct proxy of a specific biological process like axonal degeneration, which is the likely cause of the increased peripheral levels of NfL. A measure such as FA is, indeed, sensitive both to anisotropy, which is caused by the presence of elongated cell structures, like axons, and orientation dispersion, which is caused by fanning or crossing fibers [48]. Therefore, DTI cannot pinpoint the biological bases of a reduction in FA nor reflect the complexity of white matter organization. Alternative acquisition strategies have been proposed [48–50] but the estimation of an exact biologic property remains challenging despite the use of both advanced DWI sequences and modeling [51].

With these caveats in mind, the present study provides in vivo evidence of a relationship between plasma NfL levels and involvement of white matter tracts in bvFTD pathophysiology. These results advance our understanding of the link between one of the most promising fluid biomarker of neurodegeneration in FTD and the pathological process occurring in this condition and further support the application of blood-derived NfL concentration as a noninvasive biomarker for application in a clinical setting and in treatment trials.

## Supporting information

**S1 Appendix. Tracts of interest analysis: Comparison of FA values between bvFTD and healthy controls.**
(DOCX)

**S2 Appendix. Tracts of interest analysis: Association between FA and plasma NfL levels in the tracts that exhibited a reduction in FA in the bvFTD cohort.**
(DOCX)

## Acknowledgments

We would like to thank both the patients and their relative who took part to the study. Without their dedication this work would have not been possible We would also like to thank the staff of the Memory Clinics at Skåne University Hospital. The LUPROFS study on which the present work is based on was initiated 2008 by the authors CN, DvW, MLW, AS, KN, SV and EE.

## Author Contributions

**Conceptualization:** Nicola Spotorno, Christer Nilsson, Maria Landqvist Waldö, Danielle van Westen, Elisabet Englund, Henrik Zetterberg, Kaj Blennow, Santillo Alexander.

**Data curation:** Olof Lindberg, Karin Nilsson, Susanna Vestberg, Henrik Zetterberg, Jimmy Lätt, Nilsson Markus, Santillo Alexander.

**Formal analysis:** Nicola Spotorno, Olof Lindberg, Kaj Blennow.

**Funding acquisition:** Danielle van Westen, Wahlund Lars-Olof, Santillo Alexander.

**Investigation:** Christer Nilsson, Maria Landqvist Waldö, Karin Nilsson, Susanna Vestberg, Elisabet Englund, Santillo Alexander.

**Methodology:** Nicola Spotorno, Henrik Zetterberg, Kaj Blennow, Jimmy Lätt, Nilsson Markus.

**Project administration:** Santillo Alexander.

**Resources:** Wahlund Lars-Olof.

**Supervision:** Santillo Alexander.

**Visualization:** Nicola Spotorno.

**Writing – original draft:** Nicola Spotorno, Santillo Alexander.

**Writing – review & editing:** Nicola Spotorno, Olof Lindberg, Christer Nilsson, Maria Landqvist Waldö, Danielle van Westen, Karin Nilsson, Susanna Vestberg, Elisabet Englund, Henrik Zetterberg, Kaj Blennow, Jimmy Lätt, Nilsson Markus, Wahlund Lars-Olof, Santillo Alexander.

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
