## [Decision Letter · Decision Letter 0]

14 Aug 2020

Pécs, Hungary

August 13, 2020

PONE-D-20-20650

Plasma neurofilament light protein correlates with diffusion tensor imaging metrics in frontotemporal dementia

PLOS ONE

Dear Dr. Spotorno,

Thank you for submitting your manuscript to PLOS ONE. After careful consideration, we feel that it has merit but does not fully meet PLOS ONE’s publication criteria as it currently stands. Therefore, we invite you to submit a revised version of the manuscript that addresses the points raised by the Reviewers, listed below.

We look forward to receiving your revised manuscript.

Kind regards,

Joseph Najbauer, Ph.D.

Academic Editor

PLOS ONE

Journal Requirements:

3. Thank you for including your competing interests statement; "I have read the journal's policy and the authors of this manuscript have the following competing interests: HZ has served at scientific advisory boards for Denali, Roche Diagnostics, Wave, Samumed and CogRx, has given lectures in symposia sponsored by Fujirebio, Alzecure and Biogen, and is a co-founder of Brain Biomarker Solutions in Gothenburg AB (BBS), which is a part of the GU Ventures Incubator Program. . KB has served as a consultant, at advisory boards, or at data monitoring committees for Abcam, Axon, Biogen, JOMDD/Shimadzu. Julius Clinical, Lilly, MagQu, Novartis, Roche Diagnostics, and Siemens Healthineers, and is also a co-founder of Brain Biomarker Solutions in Gothenburg AB (BBS)."

Reviewers' comments:

Reviewer's Responses to Questions

**Comments to the Author**

1. Is the manuscript technically sound, and do the data support the conclusions?

Reviewer #1: Yes

Reviewer #2: Yes

2. Has the statistical analysis been performed appropriately and rigorously? 

Reviewer #1: Yes

Reviewer #2: Yes

3. Have the authors made all data underlying the findings in their manuscript fully available?

Reviewer #1: Yes

Reviewer #2: Yes

4. Is the manuscript presented in an intelligible fashion and written in standard English?

Reviewer #1: Yes

Reviewer #2: Yes

5. Review Comments to the Author

Reviewer #1: In their manuscript, Spotorno and colleagues describe the relationship between NfL levels in blood and white matter integrity as measured by DTI in a small cohort of bvFTD patients. The relevance of the study is clear and the methods and statistical analyses seem appropriate. The manuscript is generally well written. I have some minor comments:

Introduction

In the paragraph starting with ‘The present study…’, the authors mention previous studies of white matter integrity in FTD. I suggest the authors clarify which measures of white matter were used. Is this based on neuroimaging measures other than those extracted from DTI?

Methods

- The methods section mentions that the data is derived from a longitudinal FTD study. Did participants also undergo repeated MRI scans and blood sampling? If so, it would be of interest to investigate whether NfL levels also predict subsequent decline in white matter measures.

- The small sample size makes the reader question whether many patients were excluded from this study, and if so, for what reason. Can you specify this more clearly?

- Table 1: were the definite cases autopsy-proven or genetic cases?

- Table 1: besides standard deviation, can you provide the range of the interval between plasma and MRI? This is important for interpretation of the results.

Results

- In the voxel-wise TBSS analysis, it seems surprising that there are negative correlations between NfL and FA but not with other DTI measures. I suggest some comment on this in the discussion section.

Reviewer #2: In this manuscript, Spotorno and coll show their findings on a potential relationship between plasma NfL levels and involvement of white matter tracts in bvFTD. These data are very interesting and deal with a very current topic. As a further strong point, the manuscrpit is well written with a thorough data discussion. I'd just add results of a direct comparison of NFl serum levels between patients and controls. Moreover, I would be interested to know if bvFTD patients were examined for motor involvement and, if yes, I'd also add a specific correlation analysis with the CST DTI values.

6. PLOS authors have the option to publish the peer review history of their article (what does this mean?). If published, this will include your full peer review and any attached files.

Reviewer #1: No

Reviewer #2: No

---

## [Author Response · Author response to Decision Letter 0]

24 Sep 2020

Dear prof. Najbauer,

Thank you for your consideration of our work. We provide here a point-by-point description of response to reviewer comments below. For your convenience the major changes in the text of the revised manuscript are provided below in italics and highlighted in the revised manuscript.

Journal Requirements

We have checked PLOS ONE’s style requirements and edited the manuscript accordingly.

 • We note that you have indicated that data from this study are available upon request. PLOS only allows data to be available upon request if there are legal or ethical restrictions on sharing data publicly.

The justification of the restricted access to the original data has been included into the revised cover letter.

 • Update the conflict of interest statement.

The conflict of interests statement has been updated in the manuscript and reported in the revised cover letter.

 • Please amend either the abstract on the online submission form (via Edit Submission) or the abstract in the manuscript so that they are identical.

 We have updated the abstract on the online submission form.

 • Please include captions for your Supporting Information files at the end of your manuscript, and update any in-text citations to match accordingly.

The captions of the supporting information have been added to the manuscript and the in-text citations have been updated.

Page (21)

Supporting information captions

S1 Appendix: Tracts of interest analysis: Comparison of FA values between bvFTD and healthy controls

S2 Appendix: Tracts of interest analysis: Association between FA and plasma NfL levels in the tracts that exhibited a reduction in FA in the bvFTD cohort

Reviewer 1

In their manuscript, Spotorno and colleagues describe the relationship between NfL levels in blood and white matter integrity as measured by DTI in a small cohort of bvFTD patients. The relevance of the study is clear and the methods and statistical analyses seem appropriate. The manuscript is generally well written. I have some minor comments.

We thank this reviewer for this overall positive appraisal of the importance and rigor of our work.

 • Introduction: In the paragraph starting with ‘The present study…’, the authors mention previous studies of white matter integrity in FTD. I suggest the authors clarify which measures of white matter were used. Is this based on neuroimaging measures other than those extracted from DTI?

Thanks for the important question. We have updated the introduction in order to clarify which metrics have been used in the literature. The studies we referred to quantified DTI metrics or combined tensor based tractography with the quantification of DTI metrics such as FA.

Please see the changes in the Introduction section of the revised manuscript.

Page 4

“Previous studies on FTD based on the quantification of DTI metrics [20–22] as well as on a combination of tensor based tractography and quantification of DTI metrics [23] have indeed showed extensive pattern of white matter involvement including the fronto-occipital fasciculus, the superior longitudinal fasciculus, the uncinate fasciculus, the dorsal cingulum bundle, the anterior thalamic radiation, and the inferior / hippocampal portion of the cingulum bundle. We also included in our analysis tracts in which degeneration has been reported to correlate with NfL level in previous DTI studies, in particular the cortico-spinal tract [16,19].”

 • The methods section mentions that the data is derived from a longitudinal FTD study. Did participants also undergo repeated MRI scans and blood sampling? If so, it would be of interest to investigate whether NfL levels also predict subsequent decline in white matter measures.

Sadly, longitudinal MRI and blood sampling were not collected as part of the current study. We have added this information to the methods section: 

Page 5

”The LUPROFS protocol includes clinical examination, caregiver history, symptom rating, neuropsychological examination, standardized neurological examination, and, at baseline, CSF and blood sample and MRI”

 • The small sample size makes the reader question whether many patients were excluded from this study, and if so, for what reason. Can you specify this more clearly?

Thank you for this important comment. Although the sample size is limited, this is roughly what is expected based on the population size of the catchment area of the study, and the inherent difficulty of conducting research on patients with frontotemporal dementia which are, not seldom, uncooperative. Based on our own epidemiological research (Nilsson et al 2014, Plos One), in our catchments area 36 FTD individuals are expected to develop FTD during the period in which the LUPROFS study was conducted. In addition, patients were also referred from neighboring areas since our unit served as tertiary center for neurodegenerative diseases. The total number of bvFTD patients included in the LUPRUFS study was 41 (including possible, probable and definite cases). However, only 20 fulfilled the inclusion criteria of this specific study. We agree that it is critical to clarify the inclusion/exclusion procedure and we have updated the method section accordingly: 

Page 5

”The participants took part in the Lund Prospective Frontotemporal Dementia Study (LUPROFS), a longitudinal study of patients with any of the frontotemporal dementia spectrum disorders which included patients from 2009 to 2014 at the Memory Clinic of Skåne University Hospital in Lund, Sweden”

“The exclusion criteria included: > 3 lacunar strokes or any number of other type of strokes visible on MRI examination, alcohol addiction, or any other significant neurological or psychiatric comorbidity. Patients were included in the present study only if they had a clinical diagnosis of probable or definite bvFTD, underwent MRI successfully with all relevant sequences to this study, and underwent MRI within a year from blood sampling. Twenty of the 41 subjects with bvFTD included in the LUPROFS cohort, could be included in the present study.”

 • Table 1: were the definite cases autopsy-proven or genetic cases?

Thank you for the important question. The information has been added to Table 1.

 Page 6

 “6 patients fulfill definite bvFTD criteria according to neuropathological examination, 2 according to genetic examination, and one according to both.”

 • Table 1: besides standard deviation, can you provide the range of the interval between plasma and MRI? This is important for interpretation of the results.

 The interval between plasma collection and MRI scan range between 0 and 10 months. This information has been added to Table 1.

 • In the voxel-wise TBSS analysis, it seems surprising that there are negative correlations between NfL and FA but not with other DTI measures. I suggest some comment on this in the discussion section.

Thank you for highlighting this important point. It is indeed unusual that a single DTI metric is associated with a variable while the other metric are not. It has been reported that DTI metrics could be differentially sensitive to biological and pathological processes. However, we prefer to not speculate on this difference because associations between NfL levels and other DTI metrics could be hidden by the relative low statistical power of the current study and they could emerge in studies with bigger cohorts. Moreover, the tensor model is not suited to support the biological interpretation of specific metrics. We have included these considerations the discussion session of the revised manuscript.

 Page 15

“Moreover, the lack of association between plasma NfL levels and DTI metrics other than FA in the TBSS analysis is surprising when considering the extent of the association between NfL levels and FA values. A follow up study in a bigger cohort is warranted to investigate whether, with greater statistical power, associations with other metrics would emerge. Different DTI metrics could also be differently sensitive to different biological process. For example, FA appears to distinguish between progressive supranuclear palsy and Parkinson’s disease with higher level of accuracy than other DTI metrics [46], while axial and mean diffusivity seem to be the first DTI-related abnormalities to occur in Alzheimer’s disease [47]. However, one should bear in mind that, although DTI metrics are sensitive to pathological processes in white matter, they cannot be interpreted as a direct proxy of a specific biological process like axonal degeneration, which is the likely cause of the increased peripheral levels of NfL. A measure such as FA is, indeed, sensitive both to anisotropy, which is caused by the presence of elongated cell structures, like axons, and orientation dispersion, which is caused by fanning or crossing fibers [48]. Therefore, DTI cannot pinpoint the biological bases of a reduction in FA nor reflect the complexity of white matter organization. Alternative acquisition strategies have been proposed [48–50] but the estimation of an exact biologic property remains challenging despite the use of both advanced DWI sequences and modeling [51].”

Reviewer 2

In this manuscript, Spotorno and coll show their findings on a potential relationship between plasma NfL levels and involvement of white matter tracts in bvFTD. These data are very interesting and deal with a very current topic. As a further strong point, the manuscrpit is well written with a thorough data discussion.

We thank this reviewer for the positive feedback on our work.

 • I'd just add results of a direct comparison of NFl serum levels between patients and controls. Moreover, I would be interested to know if bvFTD patients were examined for motor involvement and, if yes, I'd also add a specific correlation analysis with the CST DTI values.

Although the comparison of NfL levels between patients and controls was beyond the purpose of the present study and that elevated peripheral levels of NfL in FTD have been reported in previous studies (e.g, Olsson et al., 2019, JAMA Neurol; Meeter et al., 2019, J Neurol Neurosurg Psychiatry; Landqvist Waldö et al., 2013, BMC Neurol.) we agree that the comparison between NfL in plasma between FTD patients and controls iscertainly is an important topic and the lack of such comparison in the current cohort is a limitation which have been included in the discussion of the revised manuscript.

Page 15

“ Increased peripheral NfL has now been reliably validated in FTD [8–13] and was not a purpose of the current study; however the lack of a direct comparison of plasma NfL levels between bvFTD and healthy controls is also a limitation of the present study.”

The FTD patients were examined for motor involvement but, regrettably, only in a clinical routine protocol without using any standardized measurement scale. Thus, any quantitative analysis, such as the correlational analysis suggested, cannot be performed in the present chort. However, one patient had a FTD MND diagnosis. This is explicitly mentioned in the methods section of the revised manuscript.

Page 5 

“No standardized assessment of motor functions was performed as part of the study, however, one patient had a diagnosis of FTD with motor neuron disease (FTD-MND according to the Awji criteria [25])”

---

## [Editor Report · Decision Letter 1]

12 Oct 2020

Pécs, Hungary

October 12, 2020

Plasma neurofilament light protein correlates with diffusion tensor imaging metrics in frontotemporal dementia

PONE-D-20-20650R1

Dear Dr. Spotorno,

We’re pleased to inform you that your manuscript (R1 version) has been judged scientifically suitable for publication and will be formally accepted for publication once it meets all outstanding technical requirements.

Kind regards,

Joseph Najbauer, Ph.D.

Academic Editor

PLOS ONE

Additional Editor Comments:

PLEASE NOTE: There are extra spaces inserted after reference numbers throughout the manuscript text - please correct.

---

## [Editor Report · Acceptance letter]

16 Oct 2020

PONE-D-20-20650R1 

Plasma neurofilament light protein correlates with diffusion tensor imaging metrics in frontotemporal dementia 

Dear Dr. Spotorno:

I'm pleased to inform you that your manuscript has been deemed suitable for publication in PLOS ONE. Congratulations! Your manuscript is now with our production department. 

Kind regards, 

on behalf of

Dr. Joseph Najbauer 

Academic Editor

PLOS ONE